

# Multi-label multi-class COVID-19 Arabic Twitter dataset with fine-grained misinformation and situational information annotations

Rasha Obeidat, Maram Gharaibeh, Malak Abdullah and Yara Alharahsheh

Department of Computer Science, Jordan University of Science and Technology, Irbid, Jordan

## ABSTRACT

Since the inception of the current COVID-19 pandemic, related misleading information has spread at a remarkable rate on social media, leading to serious implications for individuals and societies. Although COVID-19 looks to be ending for most places after the sharp shock of Omicron, severe new variants can emerge and cause new waves, especially if the variants can evade the insufficient immunity provided by prior infection and incomplete vaccination. Fighting the fake news that promotes vaccine hesitancy, for instance, is crucial for the success of the global vaccination programs and thus achieving herd immunity. To combat the proliferation of COVID-19-related misinformation, considerable research efforts have been and are still being dedicated to building and sharing COVID-19 misinformation detection datasets and models for Arabic and other languages. However, most of these datasets provide binary (true/false) misinformation classifications. Besides, the few studies that support multi-class misinformation classification deal with a small set of misinformation classes or mix them with situational information classes. False news stories about COVID-19 are not equal; some tend to have more sinister effects than others (*e.g.*, fake cures and false vaccine info). This suggests that identifying the sub-type of misinformation is critical for choosing the suitable action based on their level of seriousness, ranging from assigning warning labels to the susceptible post to removing the misleading post instantly. We develop comprehensive annotation guidelines in this work that define 19 fine-grained misinformation classes. Then, we release the first Arabic COVID-19-related misinformation dataset comprising about 6.7K tweets with multi-class and multi-label misinformation annotations. In addition, we release a version of the dataset to be the first Twitter Arabic dataset annotated exclusively with six different situational information classes. Identifying situational information (*e.g.*, caution, help-seeking) helps authorities or individuals understand the situation during emergencies. To confirm the validity of the collected data, we define three classification tasks and experiment with various machine learning and transformer-based classifiers to offer baseline results for future research. The experimental results indicate the quality and validity of the data and its suitability for constructing misinformation and situational information classification models. The results also demonstrate the superiority of AraBERT-COV19, a transformer-based model pretrained on COVID-19-related tweets, with micro-averaged F-scores of 81.6% and 78.8% for the multi-class misinformation and situational information classification

Corresponding authors
Rasha Obeidat,
rmobeidat@just.edu.jo
Malak Abdullah,
mabdullah@just.edu.jo

tasks, respectively. Label Powerset with linear SVC achieved the best performance among the presented methods for multi-label misinformation classification with micro-averaged F-scores of 76.69%.

## INTRODUCTION

The emergence of the Coronavirus pandemic 2019 (COVID-19) has not only posed major challenges to health systems all over the globe but also has flooded social media platforms with excessive amounts of information about the disease mixed with misleading information, which is being referred to as the global infodemic. In the Arab region, Twitter has become one of the main communication channels for people to share news and opinions about COVID-19 (*Essam & Abdo, 2021*). This has been accompanied by widespread conspiracy theories, fake cures, and anti-vaccine propaganda, making it very challenging to verify the shared information's truth. A study by *Khalifa et al. (2020)*, involving participants from nine Arab countries, shows that 33.52% of respondents believe that Twitter facilitates the spreading of coronavirus-related rumors. Another study that analyzes the magnitude of misinformation being spread on Twitter on COVID-19 reveals that 42.2% of the investigated tweets included misleading or unverifiable information (*Kouzy et al., 2020*). Further exacerbating the problem, fake news propagates on social media more rapidly than reliable news (*Dizikes, 2018*).

With the emergence of novel diseases that are not fully understood yet, health misinformation on social media can promote risky practices, posing serious threats to public health. Myriad false claims are rampant online about how to cure coronavirus and prevent its transmission without medical evidence. Whereas some claims, such as "eating garlic and drinking hot water can eliminate the virus," are harmless, some remedies can be life-threatening. Several fatalities and hospitalizations have been reported globally due to promoting the consumption of toxic substances to cure the virus (*Barua et al., 2020*; *Islam et al., 2020*). Besides, several conspiracy theories have gone viral on social media, such as the idea that 5G mobile networks are linked with the virus (*Alqurashi et al., 2021*) and that the virus is a biological weapon (*Matt Burgess, 2021*). Exposure to COVID-19 conspiracies reduces the effectiveness of official attempts to control the crisis and causes mental health issues that range from anxiety and depression to experiencing suicidal thoughts (*Su et al., 2021*).

At this stage of the pandemic, the harmful implications of COVID-19 misinformation are manifested in vaccine hesitancy. It is estimated that COVID-19 vaccines must be accepted by at least 55% of the population to obtain herd immunity (*Loomba et al., 2021*). However, achieving high public uptake of vaccines is threatened by conspiracy theories and mistrust (*Simione et al., 2021*). A relevant study (*AlTarrah et al., 2021*) that assessed

the public perceptions of the vaccine in Jordan and other Arab countries found a strong link between vaccine hesitancy and conspiratorial beliefs. For instance, 27.7% and 23.4% of respondents think that COVID-19 vaccines are intended to inject microchips into recipients and hurt fertility, respectively. Subsequently, fighting COVID-19 misinformation has become more important than ever to achieve a widespread acceptance of the vaccines for successful control of the pandemic.

Public health officials such as The World Health Organization (WHO) teamed up with leading social media companies, and fact-checking Organizations have made considerable efforts to mitigate the adverse consequences of misinformation and disseminate timely and evidence-based information to the public. However, manual fact-checking is time-consuming and may need domain expertise (*Greene & Murphy, 2021*). Thus, there is an urgent need to develop automated models that efficiently detect misinformation about COVID-19 on social media. Since the quality of misinformation detection models depends heavily on the availability of labeled data, considerable efforts have been made to create annotated datasets for COVID-19 misinformation detection from Arabic content (especially Twitter). Most of these works have focused exclusively on assigning binary labels ("true" and "false") to news stories (*Haouari et al., 2020b*; *Alqurashi et al., 2021*; *Mahlous & Al-Laith, 2021*). However, misleading stories about COVID-19 are not equal; they have varying levels of potential impacts on individuals and society (*Kabha et al., 2020*). Some tend to have more baleful than others (*e.g.*, fake cures). This suggests that identifying the sub-type of misinformation is critical for choosing the suitable action to deal with these types based on their level of seriousness.

To this end, we present and publicly share a misinformation dataset of 6.7K Arabic tweets spreading COVID-19-related claims. Our dataset is manually labeled for 19 misinformation classes representing the sub-types of the veracity (*i.e.*, falsity/truthfulness) of the information on Twitter. "Fake cure", "true cure," "fake vaccine info", and "true vaccine info" are examples of the defined misinformation classes. For instance, the tweet "تحذير! أكثر من 31 طبيبا وعالما يقولون ان لقاح كوفيد 19 ليس آمنًا وغير فعال!" (Translation: Warning! more than 31 doctors and scientists say that the COVID-19 vaccine is unsafe and ineffective!) is classified as "fake vaccine info," whereas we assigned the class "true vaccine info" to the tweet "الام ار ان اي لا يدخل النواة أصلا لكي يتلاعب في الجينات" (Translation: The mRNA does not enter the nucleus at all to manipulate the genes). In addition, we noticed that 8.29% of the tweets belong to more than one sub-type, wherefore we present a multi-label version of our dataset in which each tweet receives all of the relevant misinformation classes besides the assigned primary class. We also inspected different situational information types in Arabic tweets (*e.g.*, "caution and advice"). Since understanding situational information helps authorities or individuals understand the situation during emergencies (*Li et al., 2020a*), we present another version of our dataset annotated with situational information classes similar to the classes defined by *Li et al. (2020a)*. Besides, we design several learning tasks and experiment with various classifiers to offer baseline results for future research. Finally, we present a discussion of the potential applications of the proposed work. We believe that the misinformation detection systems that support fine-grained misinformation and situational classes are more capable of serving the specific

goals of the potentially interested parties, such as social media platforms and health professionals, compared binary classification. The contributions of this work are summarized as follows:

- We developed comprehensive misinformation annotation guidelines and released the first COVID-19-related Arabic dataset comprising about 6.7K tweets annotated with fine-grained multi-class and multi-label misinformation classes, including tweets related to COVID-19 vaccination.
- We develop comprehensive situational information annotation guidelines, and we release a version of our dataset to be the first tweets Arabic dataset annotated exclusively with situational information classes.
- We define several learning tasks over annotated data and present the experimental results of various transformers-based modes and ML classifiers using multiple feature extraction techniques.

## PRELIMINARIES

In recent years, there has been a growing interest in detecting "fake news," "misinformation," and "disinformation" in social media. However, researchers have no agreement on the definition of these overlapping terms (*Zhou & Zafarani, 2020*; *Patwa et al., 2021*). One definition of fake news adopted in various recent research works is "a claim or piece of information that is verified to be false or misleading" (*Alam et al., 2020*; *Wani et al., 2021*). Several studies further divide fake news into "misinformation" and "disinformation" based on the author's intention (*Ciampaglia et al., 2018*; *Raza & Ding, 2022*; *Kolluri & Murthy, 2021*). Whereas "misinformation" is the unintentional sharing of false information, "disinformation" is the deliberate creation and sharing of false information with malicious intent to deceive people (*Ciampaglia et al., 2018*; *Kim et al., 2020*).

Fine-grained definitions of related information disorders are also used in the literature. Examples include mal-information, news fabrications, rumors, hoaxes, satire and propaganda. Distinguishing these types of information disorders is performed based on three dimensions: the factuality of information, the intent to harm and whether the reader is implicitly aware that the information could be false or not. Mal-information is the sharing of genuine information exaggerated with the intent to inflict harm (*Alonso et al., 2021*). News fabrications are stories about non-existing events such as celebrity gossip (*Pérez-Rosas et al., 2017*). Hoaxes are intentionally crafted fake stories that aim to convince people of their veracity (*Tacchini et al., 2017*). *Satire* is a humorous news item with genuine content intending to dispense irony (*De Sarkar, Yang & Mukherjee, 2018*). Rumor is pieces of information that are unverified at the time of posting and can be verified later; in other words, they can't be opinions or feelings (*Oshikawa, Qian & Wang, 2020*). Propaganda is a biased or exaggerated story that aims to manipulate readers to advance a specific agenda (*Da San Martino et al., 2019*; *Abdullah, Altiti & Obiedat, 2022*).

Since it is extremely hard to know the intention of publishing false information about COVID-19 in this work, we follow several previous studies (*Alqurashi et al., 2021*;

*Pérez-Rosas et al., 2017*; *De Sarkar, Yang & Mukherjee, 2018*; *Hossain, 2021*) and use the term "misinformation" as an umbrella term for all false and inaccurate information. We also use the term "fake news" as an interchangeable term, especially in the related work section. For a detailed overview of the fundamental theories related to different types of information disorders, the reader is highly encouraged to refer to the survey by *Zhou & Zafarani (2020)*.

## RELATED WORK

Since the start of the COVID-19 outbreak, the study of COVID-19 misinformation has become a popular area of research. This section highlights some datasets that have been released to analyze social media data related to the COVID-19 pandemic in general. After that, we focus on the most relevant studies on misinformation detection. Then, we overview the main models and approaches proposed for COVID-19 misinformation detection from social media for the Arabic language.

### COVID-19 datasets

A considerable number of COVID-19 datasets have emerged in a short span of time after the pandemic started with multiple different goals. *Mega-COV* (*Abdul-Mageed et al., 2020*) is a billion-scale multilingual COVID dataset collected from Twitter to study a wide host of phenomena related to the pandemic. It covers 65 languages and 268 countries and has more than 169M location-tagged tweets. *Geocov19* (*Qazi, Imran & Ofli, 2020*) is a multilingual COVID-19 dataset of hundreds of millions of tweets with location information. *CORD-19* (*Wang et al., 2020*) is a growing collection of full-text scientific papers regarding COVID-19, along with their associated metadata. The Allen Institute has designed the dataset to support literature-based discoveries. *Chen, Lerman & Ferrara (2020)* collected a multilingual COVID-19 Twitter data set to understand dynamics observable in social networks during the pandemic and track the spreading of COVID-19 information on Twitter.

Several datasets for analyzing the public responses to the pandemic have been made available. *Li et al. (2020a)* studied different types of situational information about COVID-19 (*e.g.*, "caution and advice") and its propagation on Weibo. *Banda et al. (2021)* collected a large-scale English dataset of over 1.12 billion tweets related to COVID-19 and made it freely available to support a wide range of research, such as epidemiological analyses and emotional and mental responses to official measures. *Kleinberg, van der Vegt & Mozes (2020)* released a dataset of 5,000 texts regarding COVID-19 with ground truth emotional responses. *Medford et al. (2020)* built a dataset to identify the sentiment polarity and predominant emotions that appeared on Twitter during the early stages of the pandemic. *Gupta, Vishwanath & Yang (2020)* created a COVID-19 dataset comprising over 198M tweets, labeled automatically using ML and NLP techniques with latent topics, sentiment, and emotions.

Similar datasets have been released for Arabic as well. *ArCOV-19* (*Haouari et al., 2020a*) is a geo-tagged COVID-19 Twitter dataset that comprises about 2.7M tweets collected *via* Twitter search API and covers a whole year of the pandemic. *Yang et al. (2020)* created an

English and Arabic Twitter dataset of 10K tweets annotated for the task of fine-grained sentiment analysis. *Alsudias & Rayson (2020)* collected 1M Arabic tweets related to COVID-19 and clustered them using the K-means algorithm into five clusters: statistics, prayers, disease locations, advising, and advertising. *Alomari et al. (2021)* released a dataset comprising 14 million tweets from the Kingdom of Saudi Arabia (KSA) and used unsupervised Latent Dirichlet Allocation (LDA) to detect 15 government pandemic measures and public concerns.

Although these datasets have been examined and contributed to understanding social media dynamics during the pandemic, they do not provide ground truth annotation of the truthfulness of coronavirus-related information to help fight the accompanying infodemic.

## COVID-19 misinformation dataset

Many researchers have created and released COVID-19 misinformation detection datasets; the majority are with binary annotations. *Patwa et al. (2021)* manually annotated a COVID-19 fake news detection dataset containing 10,700 English social media posts and articles with binary labels ("real" and "fake"). *COVIDLIES* (*Hossain et al., 2020*) is a binary misinformation dataset containing 6,761 annotated English tweets. *Helmstetter & Paulheim (2021)* created a dataset of COVID-19 tweets for English classified into fake and non-fake tweets based on the trustworthiness of their source (*i.e.*, weak supervision). Likewise, *Zhou et al. (2020)* designed *ReCOVery*, a multimodal dataset of COVID-19 news articles and related tweets annotated weakly for veracity. Instead of fact-checking the news pieces separately, the veracity is decided collectively based on the credibility of the new sites. *CoAID* (*Cui & Lee, 2020*) is a diverse COVID-19 healthcare misinformation dataset, including fake claims crawled from reliable media outlets and automatically extracted tweets and replies related to these claims. *MM-COVID* (*Li et al., 2020b*) is a multilingual and multimodal dataset containing binary classified news content with the associated social engagements and spatial-temporal information for six different languages.

For the Arabic language, *Haouari et al. (2020b)* created *ArCOV19-Rumors*, an Arabic COVID-19 misinformation Twitter dataset containing 9.4K tweets, manually annotated into "false," "true," or "other" depending on the popular fact-checking websites. *Alqurashi et al. (2021)* created a large Arabic Twitter dataset related to the COVID-19 pandemic. The tweets are manually classified as tweets with misinformation ("false") and tweets with not-misleading information ("true"). *Elhadad, Li & Gebali (2020)* published *COVID-19-FAKES*, an English and Arabic COVID-19 tweets dataset, annotated automatically by using a community of 13 ML algorithms. *Mahlous & Al-Laith (2021)* collected more than seven million coronavirus-related Arabic tweets using trending hashtags and relied on France-Press Agency and the Saudi Anti-Rumors Authority. They manually fact-checked a sample of 2,500 tweets and annotated them into "fake" or "genuine" classes. CLEF2020 and CLEF2021 CheckThat! Lab shared tasks (*Nakov et al., 2021*; *Barrón-Cedeño et al., 2020*) released data covering several sub-tasks related to the factuality of Twitter posts on various topics, including COVID-19. For instance, one task is predicting whether a tweet is worth manual fact-checking or not (*i.e.*, the tweet is assigned to one of the classes "worth fact-checking" or "doesn't worth fact-checking"). Another task is predicting the veracity of a

news article and its topics in several languages, including Arabic (*Shahi, Struß & Mandl, 2021*).

## Class diversity

In terms of class diversity, few works provide multi-class or multi-label classification. For example, *FakeCovid* (*Shahi & Nandini, 2020*) is a multilingual cross-domain dataset of news articles for COVID-19 that are manually fact-checked from 92 different fact-checking websites and labels with 11 categories including "true," "mostly true," "partially true," "false" *etc. CMU-MisCOV19* (*Memon & Carley, 2020*) is an English multi-class misinformation tweets dataset containing fact-checked claims manually annotated with 17 fine-grained misinformation labels. To our knowledge, *Ameur & Aliane (2021)*, *Mubarak & Hassan (2020)* and *Alam et al. (2020)* are the only studies that released Arabic misinformation datasets with multi-class or multi-label annotations. These studies are the closest to our work; however, none of them offers multi-label multi-class fine-grained misinformation classification.

Specifically, *Mubarak & Hassan (2020)* manually annotated a dataset of Arabic tweets collected for COVID-19, labeled for 13 classes, including one misinformation-related class (*i.e.*, "rumor"), COVID-related topics (*e.g.*, "cure," "virus info," "governmental measures,") and situational information classes (*e.g.*, "advice," "support," *etc.*). However, this annotation scheme suffers from two problems. First, it combines rumor-related tweet identification and situational information classification in one scheme, although they serve different purposes. Second, the scheme covers important topics of the pandemic that interest people but doesn't evaluate its veracity. For example, recognizing whether this cure is fake or real is more important than only identifying that a tweet promotes a cure. Third, it doesn't support multi-label classification; thus, If there is a rumor about some governmental measure, it is unclear whether it should be labeled as a "rumor" or a "governmental measure." This work solved these issues by providing multiple versions of our dataset. The first two versions are annotated with 19 fine-grained multi-label and multi-class misinformation annotations (*e.g.*, "true treatment or cure," "fake cure," "true gov measure," *etc.*), and a third version is annotated solely with six situational information classes.

*Ameur & Aliane (2021)* created *ARACOVID19-MFH*, a dataset that combines ten independent sub-tasks (each with predefined mutually exclusive labels) in one task. For example, The subtask "factual" specify the factuality of the tweet (*i.e.*, whether the tweet describes "real" or "fake" news), the subtask "contains hate" indicates whether the label contains hate speech or not, another subtask specifies the "dialect" of the tweet, and so on. Likewise, *Alam et al. (2020)* designed dataset for a multi-label Twitter dataset for COVID-19 misinformation analysis that covers the English and Arabic languages. The annotation is performed by answering seven independent questions with predefined answers (labels) about the various aspects of the tweets, such as the worthiness of fact-checking, the need for authorities' attention, *etc*. While the suggested annotation schemes assess different independent aspects of the data, they do not present a typical multi-label setting in which tweets receive one or more mutually non-exclusive misinformation

sub-type. By analyzing COVID-19-related Twitter data, We found that the same tweet could be assigned one or more veracity sub-type based on the contents. For example, one tweet could contain "fake cure" and "fake symptoms" simultaneously. To this end, we released a version of our dataset in which tweets receive all the applicable misinformation classes. To the best of our knowledge, our dataset is the first Arabic COVID-19 dataset that supports multi-label multi-class misinformation classification. Table 1 summarizes the existing COVID-19 datasets collected for misinformation detection or situational information classification including our dataset versions, including the size, the language, the sources of samples, the annotation method, and the class granularity.

## Existing models and approaches

There has been substantial work in leveraging machine learning (ML) and deep learning (DL) to detect and classify COVID-19 misinformation. *Alsudias & Rayson (2020)* applied several ML algorithms using a sample of 2,000 tweets labeled manually for misinformation. logistic regression (LR), support vector classification (SVC), and naive Bayes (NB) classifiers are trained with two sets of features (TF-IDF and, FastText, and word2vec embeddings) to identify the rumor-related tweets with 84% accuracy achieved by LR. *Haouari et al. (2020b)* experimented with variant DL models to perform claim-level and tweet-level veracity verification using the ArCOV19-Rumors dataset they collected. The considered models are bidirectional graph convolutional network (Bi-GCN), PPC-RNN+CNN, and MARBERT and AraBERT transformer models. PPC-RNN+CNN *Liu & Wu (2018)* is a time series model that combines convolutional neural networks (CNN) and recurrent neural networks (RNN) with the propagation path classification (PPC). PPC is an algorithm that exploits user profiles to verify a tweet's genuineness. The results show that MARBERT achieved the best performance with a macro-F1 score of 74%. *Mubarak & Hassan (2020)* tested the performance of AraBERT transformer models and Support Vector Machine (SVM) classifier with several features, including word n-grams, character n-grams and Mazajak Embeddings (*Farha & Magdy, 2019*). The models are trained and tested using the multi-class *ArCorona* dataset they built. The experiments prove the superiority of AraBERT over SVM models with a macro-averaged F-score of 60.5%.

*Ameur & Aliane (2021)* evaluate the performance of five transformer models they provide as baselines for the ten independent tasks in the *ARACOVID19-MFH* dataset, including the binary misinformation detection task. The tested models are: AraBERT, mBERT, DistilBERT, AraBERT-COV19, and mBERT-COV19. The experimental results show that the last two transformer models, which have been fine-tuned using COVID-19 data, achieve the best misinformation detection performance with weighted F-scores of 95.78% and 94.91%, respectively. Likewise, *Alam et al. (2020)* demonstrate the transformer models' efficacy in performing binary and multi-class misinformation detection and fact-checking worthiness tasks. *Alqurashi et al. (2021)* applied eight different deep neural networks and ML models with multiple features, including FASTTEXT and word2ve word embeddings and word frequency, to detect COVID-19 misinformation in Arabic Twitter data. Experiments show that Extreme Gradient Boosting (XGBoost) is the best among the

**Table 1  A summary of the existing COVID-19 datasets collected for misinformation detection or situational information classification.** The summary includes the size, the language, the sources of samples, the annotation method, and the class granularity. All listed datasets are designed for misinformation detection except the ones released for situational info classification.

| Dataset | Size | Main input | Language | Classification task | Annotation method |
|---|---|---|---|---|---|
| Patwa et al. (2021) | 10.7K | Twitter, Facebook Fact-checking websites | English | Binary (fake, real) | Manually + fact-checked claims |
| COVIDLies (Hossain et al., 2020) | 6.7K | Twitter | English | Binary (fake, real) | Manually |
| Helmstetter & Paulheim (2021) | 400K | Twitter | English | Binary (fake, non-fake) | Weak supervision |
| ReCOVery (Zhou et al., 2020) | 146K | News articles Twitter | English | Binary (fake, real) | Distance Supervision |
| CoAID (Cui & Lee, 2020) | 301.1K | Social Posts User engagements News Articles | English | Binary (fake, real) | Distance Supervision |
| MM-COVID (Li et al., 2020b) | 11.1K | Twitter | Multi-lingual | Binary (fake, real) | Manually |
| ArCOV19-Rumors (Haouari et al., 2020b) | 9.4K | Twitter | Arabic | Three classes (false, true, other) | Manually |
| Alqurashi et al. (2021) | 8.7K | Twitter | Arabic | Binary (misleading not-misleading) | Manually |
| COVID-19-FAKES (Elhadad, Li & Gebali, 2020) | 0.4K | Twitter | Arabic + English | Binary (fake, real) | Automatically (13 ML algorithm) |
| Mahlous & Al-Laith (2021) (a) | 2.5K | Twitter | Arabic | Binary (fake, genuine) | Manually |
| Mahlous & Al-Laith (2021) (b) | 14.9K | Twitter | Arabic | Binary (fake, genuine) | Automatically |
| CLEF-2021 CheckThat! Lab (task 3A) (Shahi, Struß & Mandl, 2021) | 1.2K | News articles | Multi-lingual | Four classes (false, true, partially false, other) | Manually |
| FakeCovid (Shahi & Nandini, 2020) | 5.1K | Several Social media platforms | Multi-Lingual | Three classes (false, true, partially false) | Manually |
| Alsudias & Rayson (2020) | 2K | Twitter | Arabic | Three classes (false, true, unrelated) | Manually |
| CMU-MisCOV19 (Memon & Carley, 2020) | 0.5K | Twitter | English | Multi-class (17 classes) | Manually |
| Li et al. (2020a) | 3K | Weibo | English | Multi-class (eight Situational classes) | Manually |
| ArCorona (Mubarak & Hassan, 2020) | 8K | Twitter | Arabic | Multi-class (17 classes) (Mixing situational and misinformation classes) | Manually |
| AraCOVID19-MFH (Ameur & Aliane, 2021) | 10.8K | Twitter | Arabic | Ten independent tasks (each with 2–4 classes) | Manually |
| Alam et al. (2020) | 722 | Twitter | Arabic | Ten independent tasks (each with 2–3 classes) | Manually |
| Out dataset (ArCOV19-MCM) | 6.7K | Twitter | Arabic | Multi-class (19 misinformation classes) | Manually |
| Out dataset (ArCOV19-MLM) | 6.7K | Twitter | Arabic | Multi-label (19 misinformation classes) | Manually |
| Out dataset (ArCOV19-Sit) | 4.2K | Twitter | Arabic | Multi-class (Six situational classes) | Manually |

explored methods in detecting COVID-19 misinformation with an accuracy of 86.2%. Al-Yahya et al. (2021) present a comparative study of various neural networks and transformer-based language models for detecting COVID-19 misinformation from Arabic text. The experimental results demonstrate that transformer-based models outperform the neural network-based solutions with a significant improvement in the F1 score from 83.0% by the best neural network-based model( GRU) to 95.0% by the best transformer-based

model (QARiB). *Kumari (2021)* submitted the top performing system to CLEF2021 CheckThat! (shared task 3A) for fake news detection with a macro F1-score of 88.0%. They employed a BERT-based model trained with the shared task's training data and an extensive amount of additional data obtained from various fact-checking websites.

# DATASET

This section describes the steps we followed to collect and annotate our data.

## Data collection

We decided to use previously fact-checked tweets to minimize the effort needed for veracity verification and focus on assigning fine-grained misinformation and situational information class labels. We acquired most of the tweets in our dataset from available annotated binary COVID-19 misinformation datasets collected in different periods of the pandemic. We avoided the datasets labeled automatically by a community of ML classifiers. In this research, we used the following datasets:

- *ArCOV19-Rumors* (*Haouari et al., 2020b*) is an Arabic misinformation dataset of tweets manually labeled into "false," "true," or "other," we only used the tweets under "false" and "true" classes.
- A misinformation Detection sample of 2,000 tweets was manually annotated into "true," "false," and "unrelated" (*Alsudias & Rayson, 2020*). This sample was originally taken randomly by the authors from a large general-purpose unlabeled COVID-19 dataset of six million unlabeled tweets they collected using COVID-19-related keywords.
- The Arabic COVID-19 binary misinformation dataset (*Alqurashi et al., 2021*) encompasses tweets manually classified into tweets with misinformation (false) and tweets with not-misleading information (true). The annotation is performed by relying on trusted resources such as WHO and the Ministry of Health in Saudi Arabia.

We decided to use multiple datasets collected and released during different periods of the pandemic instead of relying completely on a single dataset. Our goals is to collect data that covers as many COVID-19-related subjects as possible (*e.g.*, virus properties, symptoms, prevention methods, statistics, governmental measures, *etc.*). We believe that the topics that individuals and groups vocalize about the pandemic vary with time, so we tried to obtain as comprehensive data as possible. Several studies show noticeable topic trends that change over time and in response to major events (*Ordun, Purushotham & Raff, 2020*; *Bogdanowicz & Guan, 2022*).

**Vaccine-related tweets.** Since we depend on datasets annotated and released before any COVID-19 vaccines were developed and approved, we manually collected vaccine-related tweets and verified their veracity. Specifically, We manually collected 12 common and verified claims about the COVID-19 vaccines from the Centers for Disease Control and Prevention (CDC) (https://www.cdc.gov/) and Mayo Clinic (https://www.mayoclinic.org/). Table 2 presents these claims along with their English translation. After that, we extracted a set of keywords from the claims to retrieve potentiality-related tweets. We considered keyword alternatives and repeatedly modified the search. We used Twitter API for tweet

**Table 2  Verified claims used to collect COVID-19 vaccine-related tweets and their English translations.** The first six claims are confirmed to be true, and the rest are false.

| | | |
|---|---|---|
| 1 | COVID-19 vaccine is safe and effective. | .لقاح كوفيد آمن وفعال |
| 2 | There is no evidence that COVID-19 vaccines cause infertility and miscarriage. | .لا يوجد دليل على أن لقاحات كوفيد تسبب العقم والإجهاض |
| 3 | None of the authorized and recommended COVID-19 vaccines cause people to test positive on other viral tests. | لا يتسبب أي من لقاحات كوفيد المصرح بها والموصى بها في جعل اختبار الأشخاص إيجابيًا في الاختبارات الفيروسية |
| 4 | COVID-19 vaccines cause you to be magnetized. | .تتسبب لقاحات كوفيد في أن تكون ممغنطًا |
| 5 | COVID-19 vaccines alter DNA. | .لقاحات كوفيد تغير الحمض النووي |
| 6 | COVID-19 vaccines are injected as electronic implant chips to track and monitor people. | .يتم حقن لقاحات كوفيد كرقاقات إلكترونية لتتبع الأشخاص ومراقبتهم |
| 7 | Current COVID-19 vaccines do not protect against the COVID-19 variants. | .لا تحمي لقاحات كوفيد الحالية من المتحورات |
| 8 | There are severe side effects of COVID-19 vaccines. | .هناك آثار جانبية خطيرة للقاحات كوفيد |
| 9 | The natural immunity I get from being sick with COVID-19 is better than the immunity I get from COVID-19 vaccination. | .إن المناعة الطبيعية التي أحصل عليها من الإصابة بفيروس كوفيد أفضل من المناعة التي أحصل عليها من لقاح كوفيد |
| 10 | COVID-19 vaccines are not safe because the were rapidly developed and tested. | .لقاحات كوفيد ليست آمنة لأنه تم تطويرها واختبارها بسرعة |
| 11 | The ingredients in COVID-19 vaccines are dangerous. | .المكونات الموجودة في لقاحات كوفيد خطيرة |
| 12 | I won't need to wear a mask after I get vaccinated for COVID-19. | .لن أحتاج إلى ارتداء قناع بعد أخذ اللقاح |

retrieval. After removing the irrelevant, we got 795 tweets left. We assigned "true" to the tweet that mostly paraphrases or restates a true claim or opposes a false claim (Figs. 1A and 1B). Likewise, the tweet that mostly matches a false claim or counters a true claim is labeled as a "false" tweet (Figs. 1C and 1D).

Our initial collection consists of the tweets with "true" and "false" labels from previously collected datasets after removing the duplicated tweets besides the vaccine-related tweets we've collected manually.

## Data annotation

We started by defining our misinformation and situational information annotation guidelines. We initially took a sample of the 1,000 tweets and inspected them to check the misinformation and situational information types that frequently appear. We also benefited from annotation schemes previously designed for multi-class misinformation detection (*Memon & Carley, 2020*) and situational information identification (*Li et al., 2020a*) in the English text. We have reached our fine annotation schemes after going through several class refinement iterations during which some classes have been merged, split and even deleted.

Most existing fake news detection datasets contain only two or three main misinformation classes, such as "false/true" and "fake/real/unrelated." In this work, we define a total of 19 fine-grained classes of misinformation, such as "fake cure," "true cure,"

**True claim: There is no evidence that COVID-19 vaccines cause infertility and miscarriage.**
Translation: لا يوجد دليل على أن لقاحات كوفيد تسبب العقم والإجهاض

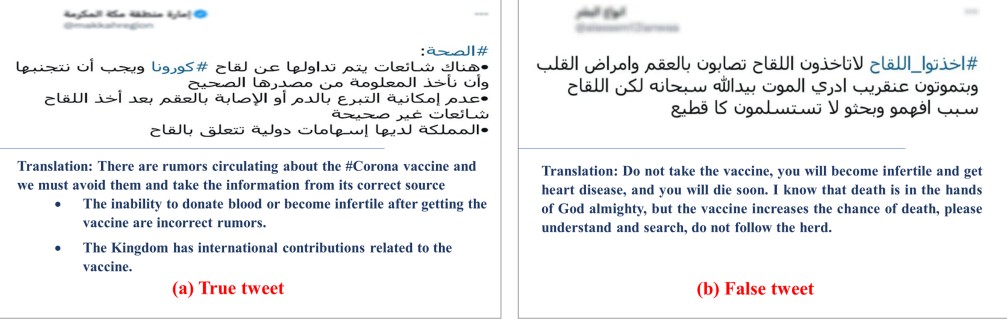

**(a) True tweet**       **(b) False tweet**

**False claim: COVID-19 vaccines cause you to be magnetized.**
Translation: تسبب لقاحات كوفيد في ان تكون ممغنطا

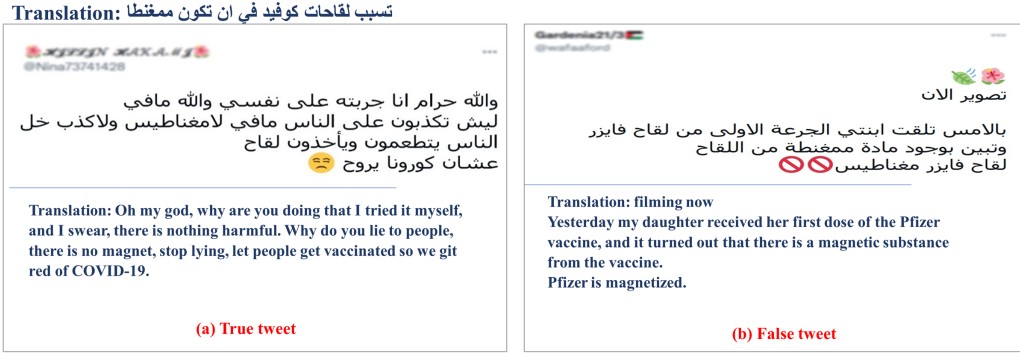

**(a) True tweet**       **(b) False tweet**

**Figure 1** Tweets in (A) and (B) are "true" and "false" tweets collected for a true claim, respectively. (C) and (D) are "true" and "false" tweets collected for a false claim, respectively (with their translations).

"fake prevention methods," "true prevention methods," "conspiracy theory," *etc*. We believe that the high-level binary (or ternary) classification does not consider the varying degrees of tweets' fact-checking worthiness or the harm-level misleading tweets could bring to society. Some types of misleading information about COVID-19 are worth more urgent fact-checking than others. Identifying the fact-checking-worthiness of Twitter data helps decide which claims should receive a higher priority from humans to perform further manual fact-checking (*Nakov et al., 2021*; *Barrón-Cedeño et al., 2020*). For instance, tweets that promote a fake cure for COVID-19 should get more attention from authorities and social media platforms and be removed urgently, whereas it could be sufficient to flag tweets with less sinister content with warning labels. Since we depend on tweets that are already fact-checked into "false" or "true," we only focus on assigning the fine-grained topic to each tweet based on the tweet content according to the definition provided in Table 3. For example, the class "true symptom" is assigned that tweets originally classified under the class "true," and at the same time, mention one or more COVID-19 symptoms. However, we had to verify the truth of some ambiguous tweets, which are not clear to which aspect of the tweet the truthiness/falseness has been assigned, as we describe shortly in the annotation challenges. "False Symptom" was one of the classes initially defined in our annotation scheme, but we excluded it since we didn't find any tweets belonging to it.

**Table 3 Description and distribution of the misinformation classes used in ArCOV19-MCM and ArCOV19-MLM.**

| Class | Description | Count in MCM | Count in MLM |
|---|---|---|---|
| True virus origin and properties | Verified information about the origin and characteristics of the virus (*e.g.*, people of all ages could be infected by COVID-19). | 221 | 221 |
| False virus origin and properties | Incorrect or unverified information about the origin and characteristics of the virus (*e.g.*, COVID-19 is killed by hot climate). | 282 | 311 |
| True spread method | Verified information about how the virus spreads between people (*e.g.*, close contact). | 116 | 137 |
| False spread method | Incorrect or unverified information about how the virus spreads among people (house flies and mosquito bites transmit COVID-19). | 242 | 269 |
| True prevention method | A verified method to prevent or mitigate the chance of being infected by COVID-19. Examples include wearing a mask, social distancing, and washing hands. | 535 | 813 |
| False prevention method | An incorrect or unverified method to prevent or mitigate the chance of being infected by COVID-19 (*e.g.*, dust prevents the spread of COVID-19). | 85 | 91 |
| True vaccine info | Correct information about COVID-19 vaccines and their effectiveness that has been verified by a competent authority such as FDA (*e.g.*, COVID-19 vaccines provide strong protection against serious illness, hospitalization and death). | 278 | 278 |
| False vaccine info | Incorrect or unverified information about COVID-19 vaccines and their effectiveness (*e.g.*, the COVID-19 vaccine causes infertility). | 442 | 443 |
| True treatment or cure | A method of treatment to reduce the symptoms or ease the pain in COVID-19 patients, such as getting plenty of sleep and liquids. This class also includes any treatment or cure method approved by competent authorities. | 68 | 69 |
| Fake treatment or cure | Any treatment or cure that is not verified by a competent authority such as WHO, FDA, or CDC or any treatment or cure proved harmful (*e.g.*, hydroxychloroquine and bleach). | 761 | 783 |
| True symptom | A symptom that has appeared in COVID-19 patients (*e.g.*, Fever, Dry cough, and Fatigue). | 112 | 180 |
| True governmental measure | It is a measure that has already been taken by a government or an official authority, such as closing country borders, shops and worship places. | 397 | 397 |
| False governmental measure | A measure that any government or official authority has not taken. | 120 | 120 |
| Conspiracy theory | A conspiracy story such as the "virus is a bioweapon" and "Bill Gates is planning to microchip the world through COVID-19 vaccines." | 312 | 320 |
| Hate and exclusion | Offensive speeches or opinions that carry feelings of hate, racism, xenophobia, or blaming others caused by the epidemic (*e.g.*, "China virus"). | 37 | 38 |
| Causing anxiety and fear | Stories that cause anxiety and spread panic among people. | 176 | 229 |
| Marketplace rumors | Rumors about products, brands, and companies related to the pandemic that could be harmful or confuse people and society. | 24 | 24 |
| Statistics | Statistics about COVID-19, including casualties, deaths, people recovered *etc.* | 370 | 409 |
| Irrelevant | The tweet cannot be classified under any of the classes above. | 2,104 | 2,104 |
| **Total** | | 6,682 | 7,235 |

Tweets that belong to "true vaccine info" and "false vaccine info" received their fine-grained misinformation classes during the collection process, as explained in the previous section. We found that 8.29% of tweets belong to more than one class, *e.g.*, a tweet

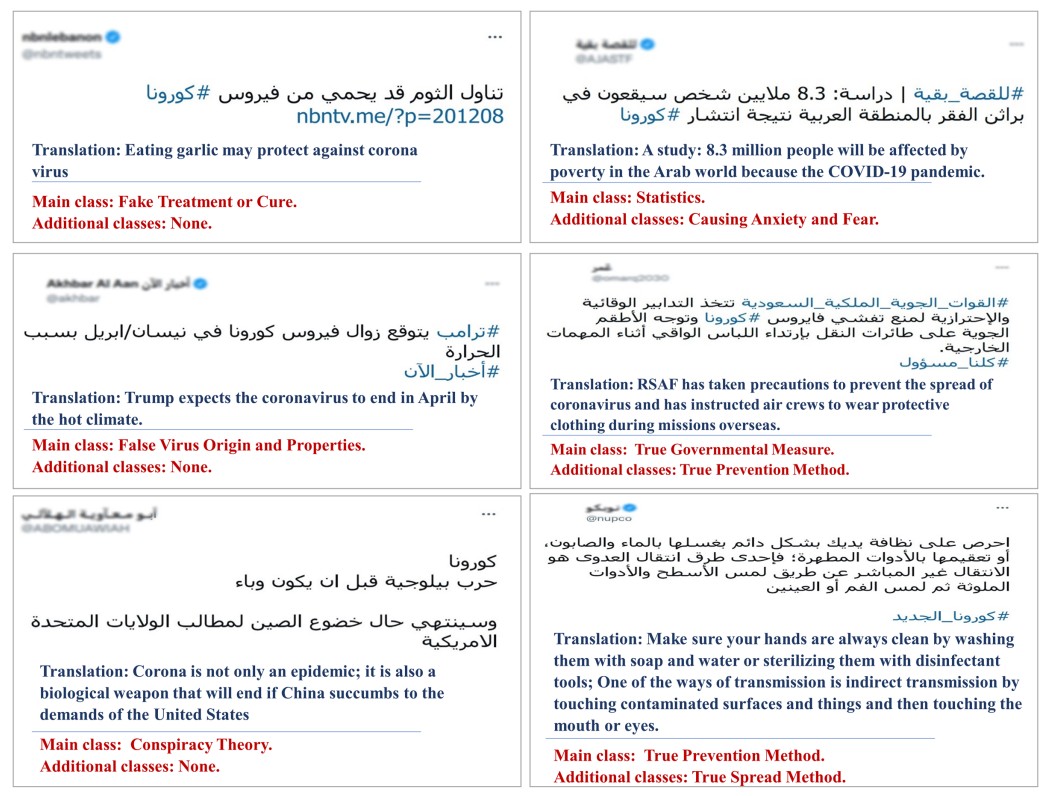

**Figure 2 Examples of tweets with multiple class labels, the examples to the left have only one primary class, while those to the right are assigned multiple labels.**

promotes a "conspiracy theory" that causes anxiety and fear (*i.e.*, belongs to the type "causing anxiety and fear"). Thus we found it necessary to allow multiple labels. Accordingly, we assigned each tweet the "primary" class to which the main point in the tweet belongs and then assigned all other applicable classes as additional class labels. As a result, we define two versions of our **Ar**abic **COV**ID-**19**-related tweet dataset (ArCOV19) with:

- **M**ulti-**C**lass **M**isinformation annotations (ArCOV19-MCM), in which each tweet is only assigned a primary class.
- **M**ulti-**L**abel **M**isinformation annotations (ArCOV19-MLM) in which each tweet might have received additional classes besides the primary class.

Figure 2 presents example tweets from our dataset with their primary and additional misinformation class labels.

**Situational information classes.** Situational information helps individuals and authorities to understand the situation during emergencies (*Li et al., 2020a*). Examples include actionable information such as asking for immediate help, offering support, advice and cautions, *etc.* Identifying situational information is important for the concerned authorities to sense the public's mood, fill information gaps with the public, and develop proper emergency response strategies (*Yan & Pedraza-Martinez, 2019*). Since situational

**Table 4 The definitions and the distribution of classes in ArCOV19-Sit.**

| Class | Description | Count |
|---|---|---|
| Caution and advice | Precautions to face the pandemic, such as washing hands, staying at home, wearing masks, avoiding travel, and encouraging compliance with precautionary measures against COVID-19 employed by authorities. | 1,222 |
| Notifications or measures | Outbreak announcements, such as the number of infections, recovery cases and deaths, and the measures taken by the concerned authorities, such as closing country borders, shops, schools and worship places and banning or restricting events. | 612 |
| Help seeking/offering | Seeking or offering support in the form of materials, money, goods, or service or asking for emotional help because of depression, fears *etc.* | 28 |
| Doubt casting and criticizing | Questioning local government officials, the Red Cross, and other related initiatives for inaction or blaming some of the public to mislead others. | 594 |
| Refute rumors | Responses against circulating rumors. | 94 |
| Non-situational | Information that can not be classified under any of the classes above. | 1,682 |
| Total | | 4,232 |

information aims to help individuals and authorities conceive the situation during major events (*e.g.*, crisis), we excluded the tweets that are originally identified as false tweets (in the original datasets) from our dataset version that is annotated with situational classes, as we believe that it is illogical relying on data with misinformation to understanding the situation during major events. We define six types of situational information and assign one situational information type to each tweet to build (ArCOV19-Sit), a version of **ArCOV19** with multi-class **Sit**uational annotations. These classes, along with their definitions and counts, are presented in Table 4.

   **Annotation challenges**. We faced the following challenges during the annotation process.

1. We initially started with two classes related to the COVID-19 statistics in our tweet collection, "true statistics" and "false statistics." However, we found that statistics change drastically over time, so we combined them into one class called "statistics."

2. As aforementioned, we use previously fact-checked tweets to minimize the effort needed to check their veracity. However, By looking at tweets containing information about the disease, such as cures, treatments, symptoms, vaccines, *etc.*, we found that some of these tweets also tell other types of stories about other entities involved, such as the person who promoted this fake cure or warned from taking that vaccine. We found that the original binary misinformation class is sometimes assigned based on whether that "other story" happened or not instead of assigning the misinformation class based on the falseness or trueness of the scientific information in the tweet. For example, by looking at the tweet presented in Fig. 3A, Donald Trump's proclaimed a prescription to treat COVID-19 patients by injecting them with antiseptics. The binary class assigned to this tweet is "True" because Trump did suggest that, but the fine-grained misinformation class we gave is "false treatment or cure." This forced us to not depend on the assigned binary class and manually verify the misinformation of all multi-story tweets that involve scientific information about the virus from credible resources, including WHO, CDC, Mayo Clinic and the FDA. We added a verification link (a link to the page we used

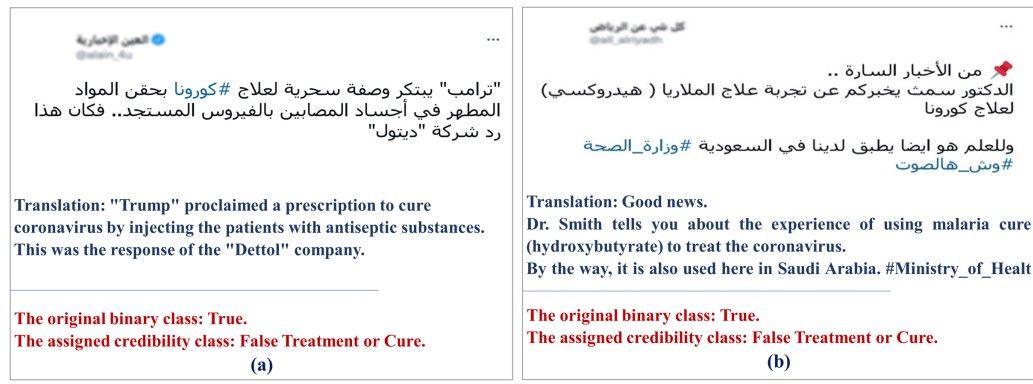

**Figure 3** Examples of tweets that posed additional challenges during data annotation.

for verification) to each of these tweets in our data. We removed any "multi-story" tweet that we could not verify the truthfulness of from our dataset.

3. At the beginning of the pandemic, some drugs, such as hydroxychloroquine that is used to treat malaria, were approved by FDA for emergency use to treat COVID-19. Later, FDA revoked this emergency use authorization (*Office of the Commissioner, 2022*). Since we depend on Twitter datasets that have been released relatively at the early stages of the pandemics, we couldn't rely on the original binary class label to classify the cure-and-treatment tweets into "false treatment or cure" and "true treatment or cure" as it might have changed as with tweet in Fig. 3B. To this end, we manually performed this classification of all of the cure-and-treatment tweets by referring to the credible resources mentioned earlier.

**Annotation quality**. We randomly selected 1.5K tweets along with their multi-label and multi-class misinformation classes (*i.e.*, from ArCOV19-MCM and ArCOV19-MLM). Then, we recruited a second annotator, who has no access to the labels by the first annotator, to perform a second round of annotation. The inter-annotator agreement of the misinformation annotations in ArCOV19-MCM is 97.83% using Cohen's kappa coefficient. The inter-annotator agreement of the misinformation annotations in ArCOV19-MLM is 95.78, using Krippendorff's alpha reliability coefficient (*Bhowmick, Basu & Mitra, 2008*). As for situational information classes, we excluded the tweets that didn't receive situational classes from the 1,500 examples mentioned above (*i.e.*, the tweets that were originally classified as false tweets in the binary misinformation datasets from which we collect our data). The inter-annotator agreement of the situational annotations in ArCOV19-Sit is 96.79 using Cohen's kappa coefficient.

The high annotation agreements of the three dataset versions prove the annotations' quality, given that some classes are more semantically related than others. The main reason for annotations divergence in ArCOV19-MLM is the subjectivity in assigning one of the classes of the applicable classes to the tweets that belong to more than one class, which assures the importance of dealing with misinformation classification as a multi-label

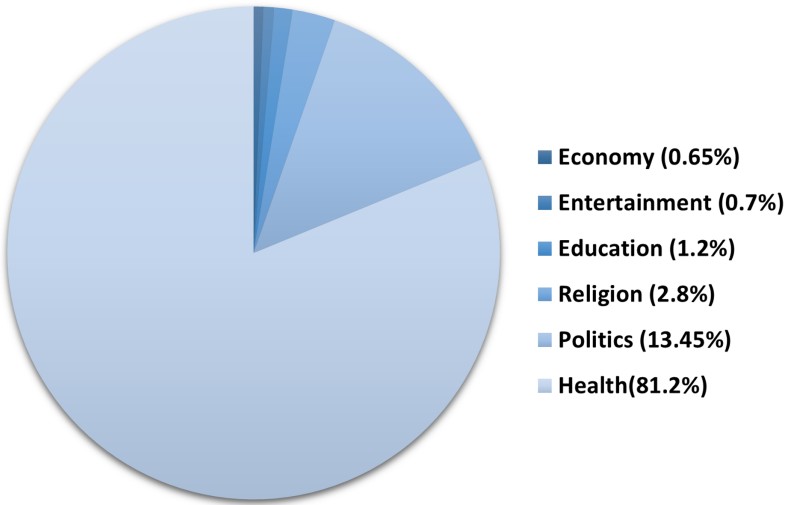

**Figure 4  Topics distribution in ArCOV19-MCM.**   

classification problem. The testing sets of ArCOV19-MCM, ArCOV19-MLM, and ArCOV19-Sit are sampled from the tweets that received full agreement between the two annotators to guarantee high-quality testing data.

## Data analysis and discussion

In this section, we performed an exploratory analysis of our dataset. The versions ArCOV19-MCM and ArCOV19-MLM contain 6,681 tweets annotated with multi-class and multi-labeled misinformation classes. ArCOV19-Sit has 4,236 tweets. The label cardinality of ArCOV19-MLM is 1.08. Label cardinality is the average number of labels per tweet (*Nam et al., 2014*). It's an indication of how hard the classification problem is. The low cardinality denotes a less-complex label space (*Sorower, 2010*). Our dataset has considerably low cardinality, which suggests that the classification task is not very difficult.

   **Topic analysis.** The global outbreak of COVID-19 affected almost all aspects of life. To assess the impact of COVID-19 on the different sectors, we annotated the 2,000 tweets from ArCOV19-MCM with the general topic they belong to, including health, education, economy, politics, religion and entertainment. Figure 4 shows that health is the most frequent topic, followed by politics. We believe that the entertainment sector received less attention from social media users not because it has not been affected but because it has become a less-vital matter compared to health.

   **Class distribution.** Figures 5A and 5B represent the class distributions of the training and the testing sets in ArCOV19-MCM, respectively. Both sets have imbalanced class distributions. The class sizes are consistent between the testing and the training sets. The most common class in the training set is the class "irrelevant," with 1,884 tweets, followed by "fake treatment or cure," with 639 tweets. The least populated class is "true treatment or cure." A plausible explanation is is that COVID-19 is relatively new; hence there is limited evidence regarding specific antivirals that may work against it (*Singh & de Wit, 2022*). The

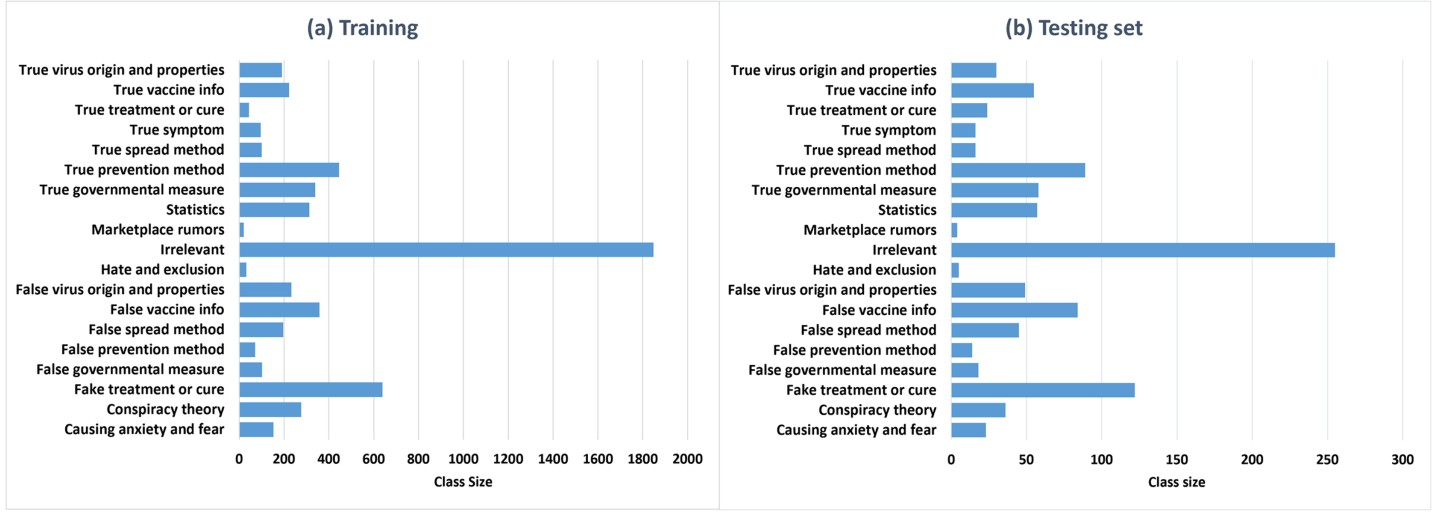

**Figure 5** The class distributions of the training (A) and the testing (B) sets of ArCOV19-MCM.

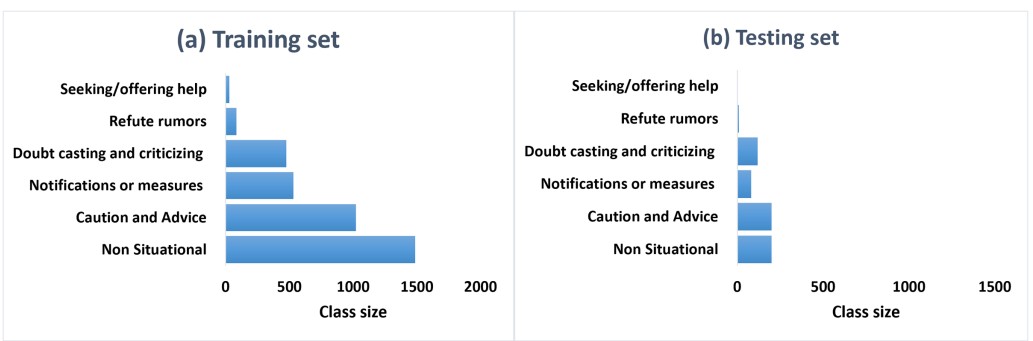

**Figure 6** The class distributions of the training (A) and the testing (B) sets of ArCOV19-Sit.

class distributions of the training and the testing sets of ArCOV19-Sit are shown in Fig. 6. The distributions are heavily skewed. The smallest class is "Help seeking/offering," and the "non-situational" class is the highest populated. The training set's most commonly populated situational information class is "caution and advice," followed by "notification and measures".

**Tweet length.** The length of the tweets in the dataset is an important property as it influences the dimensionality of the text's feature vector that represents the input to the ML model. For example, TF-IDF is usually too sparse for short text. As shown in Fig. 7, our tweets have a length range from 3 to 87 tokens, most of them with 20 to 50 tokens. We also observed no variations in tweet lengths among different classes.

**Word cloud.** Figure 8 provides the word clouds for the six misinformation classes along with their translations by picking the top 200 words in each class. Several interesting insights can be observed from these word clouds. For instance, some of the frequent words that appear in the class "true vaccine info" are آمن, الصحة and تقديم that are

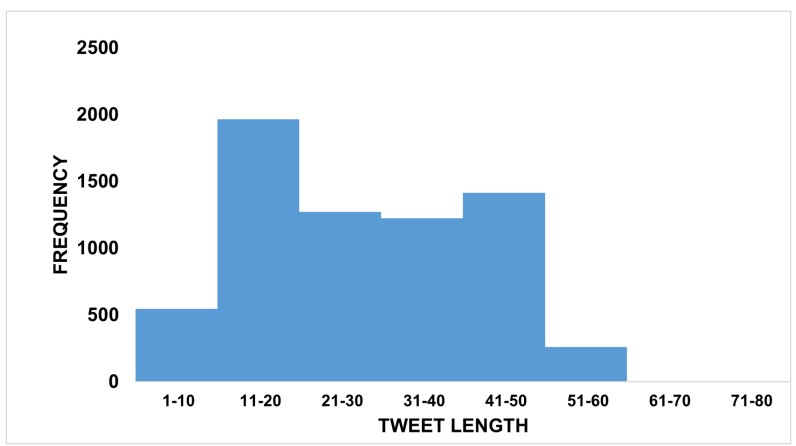

**Figure 7  Tweet length distributions in ArCOV19-MCM and ArCOV19-MLM.**

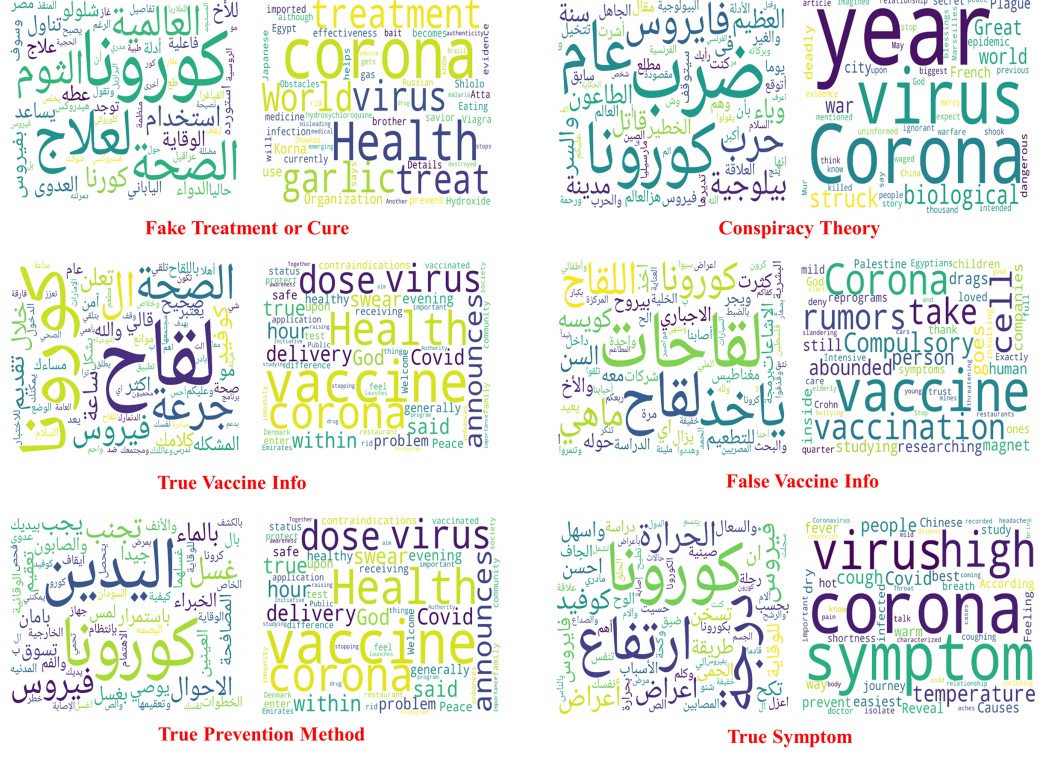

**Figure 8  Word clouds of six of the misinformation classes along with their translated clouds.**

translated to *safe*, *health*, *vaccine* and *delivery*. These positive words encourage vaccine intake. However, by looking at the frequent words of the class "False vaccine info" such as اجباري، اشاعات، مغناطيس  and that are translated into *magnet*, *rumors*, and *compulsory*. These words increase vaccine hesitancy. Likewise, the tweets under the class "true prevention methods" have a frequent appearance of words that encourage personal

hygiene, such as غسل and اليدين (translated to *wash* and *hand*). In the class "conspiracy theory", the most frequently used words are بيولوجية and حرب (translated to *biological* and *war*), promoting one of the most widespread COVID-19-related conspiracy theories since the start of the pandemic (*Matt Burgess, 2021*).

## BASELINES AND EVALUATION

### Classification methods

This section explores the performance of several Ml and DL methods to perform multi-class misinformation classification, multi-label misinformation classification and multi-class situational information classification, representing baselines for future research using our data.

#### Multi-class classification

We performed classification into the misinformation and the situational classes by training several ML and transformer-based classifiers using ArCOV19-MCM and ArCOV19-Sit. The following is an overview of the classification models used in our experiments. We describe the features we utilize to transform the raw tweets into feature vectors representing the ML classifiers' input.

**Features.** In this study, we use three popular features to convert all tweets to numerical vectors before being fed to the ML algorithms. (1) Term Frequency/Inverse Document Frequency (TF-IDF) is a numerical statistic measure computed by multiplying two metrics to assign a weight of importance to each term in the tweet. First, Term Frequency (TF) measures the importance of a term in a tweet (document). Second, IDF ($t$) is a weight that indicates how frequently a term appears in the dataset. The terms that are common in the dataset receive low IDF scores. (2) N-Grams are contiguous sequences of N elements in a given text. An element in an N-gram could be a character, word, or even a sentence and is decided based on the task. It is obtained by sliding a window of size N over the text. An N-gram of size one is referred to as a unigram; 2-gram is called a bigram, and 3-gram is known as a trigram. We experimented with character-level bigrams. (3) Stylistic features are a type of features that help to discover different authoring and writing styles for the authors of fake news *vs* authors of legitimate news (*Hussein, 2019*). We extracted 14 hand-crafted stylistic features, including the number of words, characters, hashtags, user mentions (@), URLs, links, emojis, ellipses, digits, commas, period, punctuation marks, exclamation marks, percentage marks and question marks.

**Machine learning algorithms.** We employ three well-known ML algorithms to perform multi-class misinformation classification: SVM (*Pisner & Schnyer, 2020*), random forest (RF) (*Cutler, Cutler & Stevens, 2012*), and XGBoost (*Chen & Guestrin, 2016*), as they have proven to be successful in various Arabic text classification tasks including sentiment analysis (*Abooraig et al., 2018*), and hate speech detection (*Aljarah et al., 2020*).

**Transformer-based models.** Transformers (*Vaswani et al., 2017*) are neural network architecture that applies the mechanism of self-attention to selectively weigh the significance of each part of the input text. They have been pretrained on huge unsupervised general corpora. Pretrained transformer-based models such as BERT (*Devlin et al., 2018*)

and post-BERT models like RoBERTa (*Liu et al., 2019*) and ALBERT (*Lan et al., 2019*) have achieved state-of-the-art performance for several NLP tasks classification and generation tasks. In this work, we use five transformer models:

- **AraBERT** (https://huggingface.co/aubmindlab/bert-base-arabertv02) is a BERT model pretrained on 200 million Arabic MSA sentences gathered from various resources based on Google's BERT-Base configuration (*Antoun, Baly & Hajj, 2020*).
- **AraBERT-COV19** (https://huggingface.co/moha/arabertc19) is a version of AraBERT pretrained using an Arabic COVID-19 Twitter dataset (*Ameur & Aliane, 2021*). The dataset consists of 1.5 million raw multi-dialect Arabic tweets about the COVID-19 pandemic.
- **AraBERTv02-Twitter** (https://huggingface.co/aubmindlab/bert-base-arabertv02-twitter) is a transformer-based model obtained by continuing the pretraining AraBERT using 60M multi-dialect Arabic tweets.
- **MARBERTv02** (https://huggingface.co/UBC-NLP/MARBERTv2) MARBERT is a transformer-based model that exploits large-to-massive scale multi-dialect datasets. It has been evaluated on several NLP tasks, including sentiment analysis, Question Answering and dialect identification. MARBERTv02 is MARBERT that has been further fine-tuned using additional data with longer sequences to handle the task that needs to capture longer contexts (*Abdul-Mageed, Elmadany & Nagoudi, 2021*).
- **XLM-R** (https://huggingface.co/cardiffnlp/twitter-xlm-roberta-base) is a transformer-based masked language model trained jointly on one hundred languages, including Arabic, to improve the cross-lingual understanding tasks (*Barbieri, Espinosa-Anke & Camacho-Collados, 2022*; *Conneau et al., 2020*).

### *Multi-label classification*

Unlike multi-class classification, in which each instance is associated with an exactly single class label, multi-label classification is a supervised learning problem in which the data instance can simultaneously belong to more than one class. Multi-label classification methods can be roughly divided into problem transformation and algorithm adaptation. The problem transformation method involves transforming a multi-label problem into one or more independent binary or multi-class classification tasks. The final multi-label prediction for a new instance is determined by aggregating the classification results from the individual classifiers. On the other hand, algorithm adaptation methods extend traditional learning algorithms to handle multiple labels directly (*Tsoumakas & Katakis, 2007*). We use two problem transformation methods: binary relevance (BR) (*Boutell et al., 2004*) and label powerset (LP) (*Tsoumakas, Katakis & Vlahavas, 2009*). LinearSVC and RF are adopted as base classifiers.

### Experimental setup
### *Evaluation metric and model selection*

Following prior works, We evaluate the performance of our baseline models by reporting macro-averaged and micro-averaged Precision (P), Recall (R) and F1 scores in addition to

**Table 5  The optimal hyperparameters of the ML algorithms used in multi-class misinformation and situational information classification.**

| Classifier | Hyper-parameters | Misinformation classification | Situational classification |
|---|---|---|---|
| SVM (linear kernel) | C | 1 | 1 |
| | Gamma | 1 | 1 |
| RF (entropy criterion) | n_estimators | 50 | 200 |
| | max_depth | 100 | 500 |
| | max_features | sqrt | sqrt |
| XGBoost | max_depth | 10 | 10 |
| | n_estimators | 100 | 100 |
| | random_state | 42 | 100 |
| | eta | 0.3 | 0.3 |
| | subsample | 1 | 1 |

Accuracy. We use the micro-averaged F1 score as the primary metric for comparison. In addition, we employ the Jaccard score as an additional evaluation metric of multi-label classification. To evaluate the performance of the considered multi-class and multi-label misinformation classification models, we sampled 1,000 tweets out of the data that the annotators agreed upon as testing sets of dataset versions ArCOV19-MCM and ArCOV19-MLM. The goal is to maintain the same division across all models to allow a fair comparison. The testing set of ArCOV19-Sit consists of 608 tweets that resulted from removing the tweets that were originally annotated as false (in the source binary datasets) from the 1,000-tweet sample. As we mentioned earlier, some tweets in textbfArCOV19-MLM received additional classes besides the primary class. It is worth noting that we don't distinguish the primary class label from the additional class labels during modeling.

### Hyper-parameter tuning

This section presents hyper-parameter tuning performed to the learning techniques considered in this study. We applied the grid search (*Liashchynskyi & Liashchynskyi, 2019*) to identify the optimal hyper-parameters for the ML algorithms and tried selected values to tune the transformer-based classifiers. We use the training set of each version to fit the models. The hyper-parameters are tuned using 10-fold cross-validation. The models that achieve the best cross-validation micro-averaged F-scores are applied to the testing set to report the final results.

Table 5 presents the optimal hyper-parameters of SVM, RF, and XGboost used in the multi-class misinformation and situational information classification tasks. We use the SVM linear kernel (LinearSVC) with the One-*vs*-One approach implementation provided by scikit-learn (https://scikit-learn.org/stable/). Regarding the BERT models, including AraBERT, mBERT and AraBERT COV19, the AdamW optimizer is used with a learning rate of 4e−5, the batch size is set to 8, and the maximum sequence length is 128. We used early stopping to tune the number of epochs and found that three epochs yielded the best performance. We adopted the default hyperparameters of LinearSVC and RF used for multi-label misinformation classification.

**Table 6 Performance of the multi-class misinformation classification models trained and evaluated using ArCOV19-MCM.** Boldface represents the performance scores of the best ML classifier and the best transformer-based for each metric.

| Classifier | Accuracy | Macro-P | Macro-R | Macro-F1 | Micro-F1 |
|---|---|---|---|---|---|
| SVM | **77.3%** | 80.55% | **66.86%** | **71.38%** | **77.3%** |
| RF | 56.1% | **85.68%** | 41.74% | 48.55% | 56.1% |
| XGBoost | 72.39% | 79.01% | 61.78% | 67.19% | 72.39% |
| AraBERT | 80.3% | 83.46% | 73.79% | 76.24% | 80.3% |
| AraBERT-COV19 | **81.6%** | 80.69% | **77.96%** | **78.76%** | **81.6%** |
| AraBERTv02-Twitter | 81.1% | **81.27%** | 75.95% | 77.02% | 81.1% |
| MARBERTv2 | 80.1% | 76.39% | 72.98% | 73.15% | 80.1% |
| XLM-R | 79.1% | 75.32% | 71.16% | 71.21% | 79.1% |

## Results and discussion

This work defines three classification tasks: multi-class misinformation classification, multi-label misinformation classification, and multi-class situational information classification. Several Ml classifiers and transformers-baseline models are trained and tested for each classification task representing baselines for future research on our data. This section reports the experimental results of these classifiers for each classification task and discusses reported performance.

### Multi-class misinformation classification

We applied the ML classifiers, XGBoost, SVM and RF, and five transformer-based pretrained models: AraBERT, AraBERT-COV19, AraBERTv02-Twitter, MARBERTv2 and XLM-R to perform multi-class misinformation classification. All models are trained using the training set of ArCOV19-MCM and evaluated using the corresponding testing set. The ML classifier was trained using a combination of n-grams and TF-IDF stylistic features. We extracted the word-level unigrams and bigrams and char-level bigrams and trigrams. Then, we computed the TF-IDF of each type. The final vector representing each tweet is a concatenation of these TF-IDF vectors and the stylistic features after being normalized to have the same scale as the TF-IDF scores. These vectors represent the input of the ML models. The values of accuracy, macro-averaged and micro-averaged precision, recall, and F1 scores are repaired. All the experimental results of this task are summarized in Table 6. It is worth mentioning that the accuracy and the micro-averaged F1 scores for multi-class classification problems are mathematically equivalent; thus, they have identical values. The best results are highlighted in bold font.

**Results of the ML classifiers.** Several observations and insights can be drawn from these results. First, among the compared ML algorithms, SVM achieved the best accuracy of 77.3%, macro-averaged recall of 66.86% and macro-averaged F1 score of 71.38%, followed by XGBoost. Second, the micro-averaged F1 scores are consistently higher than the macro-average F1 scores of the three models. This is because the dataset ArCOV19-MCM has imbalanced class distribution. For example, 31.6% of the instances belong to the class "irrelevant," and the rest are distributed unevenly over the remaining classes (See Fig. 5).

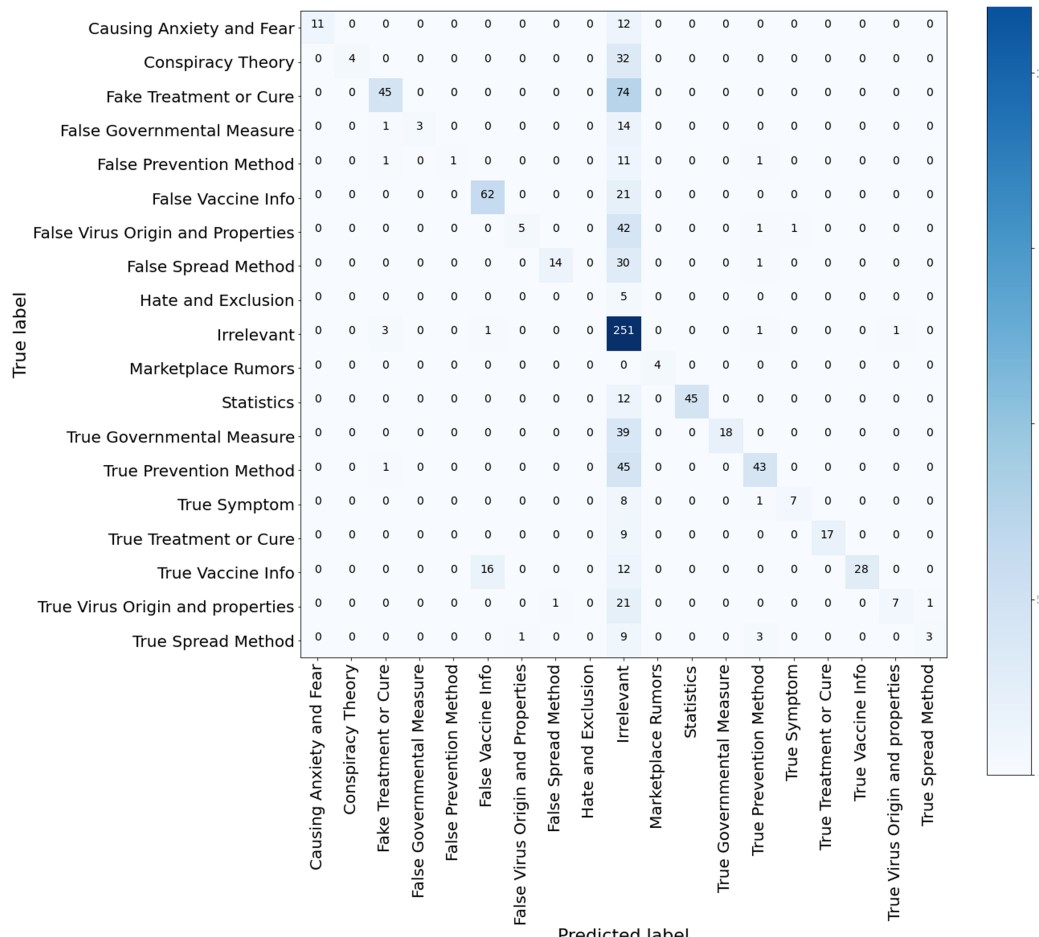

**Figure 9** **Confusion matrix RF model trained for multi-class misinformation classification.**

We thought in the beginning that it would be better if we excluded the class "irrelevant" from the data used to establish the baselines. Then, we decided to keep it and see which model is more efficient in dealing with this challenging yet common problem. Third, RF has the worst recall of 41.74% and the highest precision compared to all classifiers, including the Bert models, probably because RF is very sensitive to imbalanced data. Since the RF algorithm samples a subset from the training set to train every decision tree, these subsets will contain fewer tweets from the minority classes. Thus, RF could fail to recognize these tweets amplifying the skewed data effect (*Chen, Liaw & Breiman, 2004*). By observing the confusion matrix of the RF classifiers depicted in Fig. 9, most of the tweets in the minority classes were categorized incorrectly under the class "irrelevant." The same reason also explains the RF's high precision score, as most of the tweets are classified under the majority classes, irrelevant, and the percentage of the predicted majority-classes predictions that are correctly classified is high. Although SVM and XGboost significantly outperform RF, they also suffer from the uneven distribution of the classes, manifested by the differences between the macro-averaged and the micro-averaged F1 scores of these two

classifiers. These findings suggest that there is room for improvement by employing techniques that deal with imbalanced class distributions (*He & Garcia, 2009*; *Fernández et al., 2018*), or by collecting more data to correct the variations in class sizes. This is particularly important to attain the main purpose of defining fine-grained misinformation classes. Another observation can be obtained from RF's Confusion matrix in Fig. 9 that the model mixed up the tweets that belong to the classes "True vaccine info" and "False vaccine info," probably because the tweets in these two classes use similar vocabulary (as shown in Fig. 8), limiting the ability of the ML models to differentiate them. A possible solution is by augmenting our data with features extracted from Author profiles and user engagements instead of depending solely on the tweet's contexts.

**Results of the transformer-based models.** Table 6 reveals that AraBERT-COV19 followed by AraBERTv02-Twitter outperforms all models, including the ML methods, with an accuracy of 83.20%, macro-precision of 81.82%, macro-recall of 78.71% and a macro-F1 score of 79.52%. We think that AraBERT-COV19 surpassed AraBERT since it is tuned using 1.5M multi-dialect Arabic tweets related to COVID-19, making it familiar with the Arabic COVID-19 multi-dialect vocabulary. Also, since AraBERT-COV19 is trained on a Twitter dataset, it is more suitable to represent tweets in which text can be misspelled, abbreviated and has irregular syntax. In contrast, the AraBERT was mostly pretrained in Modern Standard Arabic (MSA), thus less capable of dealing with multi-dialectal Arabic than AraBERT COV19. Following AraBERT-COV19, AraBERTv02-Twitter also performs comparably well since it is a version of AraBERT obtained by continuing the pretraining AraBERT using 60M multi-dialect Arabic tweets. By observing the confusion matrix of AraBERT-COV19 in Fig. 10, AraBERT-COV19 is remarkably more capable of dealing with the imbalanced data and capturing the semantic differences between the defined fine-grained misinformation classes, confirming the efficiency of the transformers and their ability to transfer knowledge between domains and proving the importance of pretraining the BERT-based models on COVID-19-related unsupervised data.

The least-performing transformer-based method is XLM-R. A plausible explanation is that XLM-R is pretrained using multi-lingual data and usually outperformed by the monolingual models pretrained with large language-specific datasets and rich vocabularies (*Virtanen et al., 2019*; *Alammary, 2022*). MARBERT achieved comparable performance to AraBERT in the micro-averaged F1-score but suffered from a performance gap in terms of the macro-averaged F-scores, although MARBERT is pretrained with larger data. A systematic review of BERT Models for various Arabic text classification problems (*Alammary, 2022*) shows that AraBERT outperformed MARBERT in several tasks and *vice versa*. It also shows that a large pretraining *corpus* does not necessarily guarantee better performance. It would be important to study different BERT-based models by connecting their performance with the distribution of the classes in the considered tasks.

### Multi-class situational information classification

Table 7 presents the experimental results of multi-class situational classification models, trained and evaluated using the ArCOV19-Sit dataset. We employ the same baseline

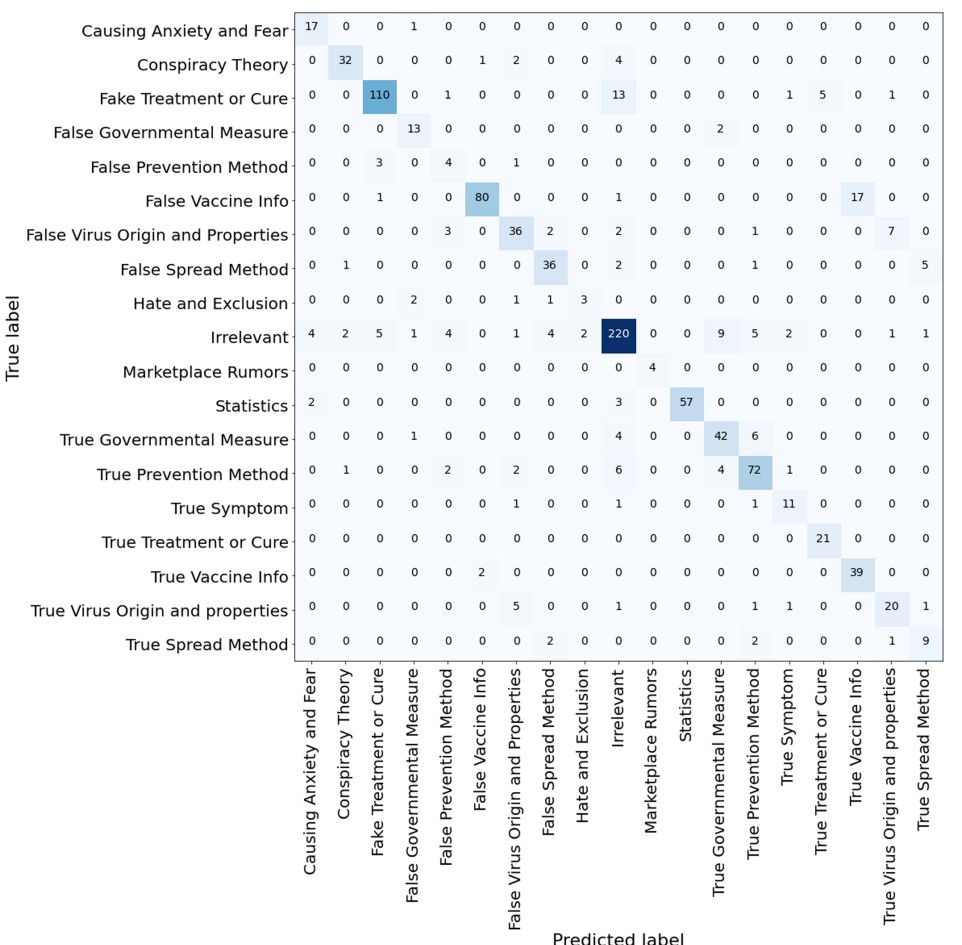

**Figure 10 Confusion matrix AraBERT COV19 model trained for multi-class misinformation classification.**

**Table 7 Performance of the multi-class situational information classification models trained and evaluated using ArCOV19-Sit.** Boldface represents the performance scores of the best ML classifier and the best transformer-based for each metric.

| Classifier | Accuracy | Macro-P | Macro-R | Macro-F1 | Micro-F1 |
|---|---|---|---|---|---|
| SVM | **73.39%** | 54.38% | 52.79% | **53.07%** | **73.39%** |
| RF | 66.83% | **59.16%** | 44.18% | 46.78% | 66.83% |
| XGBoost | 73.23% | 56.54% | 49.71% | 51.47% | 73.23% |
| AraBERT | 77.5% | 67.56% | 58.79% | 61.24% | 77.5% |
| AraBERT-COV19 | 78.81% | **73.96%** | **77.3%** | **73.45%** | 78.81% |
| AraBERTv02-Twitter | 77.99% | 64.66% | 76.62% | 66.58% | 77.99% |
| MARBERT | **79.31%** | 58.9% | 58.11% | **58.28%** | **79.31%** |
| XLM-R | 77.83% | 61.19% | 58.57% | **59.37%** | 77.83% |

models, features and evaluation measures used in the multi-class misinformation classification tasks. These results of ML algorithms are in concordance with the results presented in the previous section. Among the ML classifiers, SVM achieves superior

**Table 8 Performance of the multi-label misinformation classification models trained and evaluated using ArCOV19-MLM.** Boldface represents the best performance scores for each metric.

| Classifier | Binary Relevance (BR) | | | Label Powerset(LP) | | |
|---|---|---|---|---|---|---|
| | Jaccard | Macro-F1 | Micro-F1 | Jaccard | Macro-F1 | Micro-F1 |
| RF | 39.35% | 48.61% | 55.25% | 62.65% | 55.4% | 62.54% |
| Linear SVC | 60.29% | 67.84% | 75.23% | **76.3%** | **68.89%** | **76.69%** |

performance to XGBoost and RF with an accuracy of 73.39%, macro recall of 52.79%, and macro-F1 score of 53.07%. Comparing various pretrained models, AraBERT-COV19 outperformed all others in macro-averaged precision, recall, and F1 score, while MARBERT achieved the best micro-averaged F1 score. Based on the results of this section and the previous section, we hypothesize that MARBERT is more sensitive to the imbalanced class distribution, emphasizing the need to study the relationship between different BERT pretrained models and the label space. Once again, XLM-R performed the worst in most of the cases.

### Multi-label misinformation classification

This section presents the experimental result of the BR and LP multi-label classification algorithm trained using the ArCOV19-MLM dataset by utilizing LinearSVC and RF as base classifiers. The macro-averaged and micro-averaged F1 scores and Jaccard scores are reported in Table 8. Notably, the linear SVC outperformed RF as a base classifier in BR and LP, and this is consistent with our previous findings that explain why RF is more sensitive to the skewed distribution of the data. Table 8 also shows that LP outperforms BR using all measures regardless of the base classifier. BR decomposes the multi-label learning task into 19 independent binary learning tasks, one class label. Each binary classifier outputs "true" if the tweet belongs to that class or "false" if it doesn't. Then binary classifier votes individually to obtain the final predictions. In other words, the predicted label set consists of the class labels that the corresponding binary classifiers output "true." On the other hand, the LP approach transforms a multi-label problem into a single-label multi-class problem and applies a multi-class classification approach to solve the problem. The classes include all class combinations that appeared in the training set. By observing the class distribution of ArCOV19-MLM presented in Table 3, we can see that both types of transformation performed by LP and BR increase the class imbalance. However, the approach that is used by BR transformation causes a more severe data imbalance not only for the minor classes but also for the moderately-populated classes. This suggests applying adaptation-based multi-label classification approaches which don't include data transformation could improve the performance for this task, which we leave for future work.

### Discussion of potential applications

In light of the experimental results that confirm the practical utility of our data, several parties can potentially benefit from the presented work, including, but not limited to, social media platforms, health professionals, public authorities and decision-makers. Assigning a

specific misinformation sub-type to COVID-19-related social media posts can be employed by the social media platforms to direct the information to the most interested audience. It also helps assess the degree of harm misinformation can bring people and take the appropriate measures accordingly. Possible measures could range from assigning warning labels to the susceptible post to removing posts with malicious content or even suspending the account that shares the misleading posts entirely (*Levush, 2019*; *Alam et al., 2020*). Health professionals can utilize the suggested models to identify the health-related misinformation among the existing diverse misinformation themes, which is considered the first step in fighting their proliferation. Fine-grained misinformation and situational information can be exploited by the public authorities and decision-makers to understand the situation and the public mood during the pandemic peak, assisting in making better decisions to guide people, support them and save their lives.

## CONCLUSION

The outbreak of COVID-19 has accompanied the massive spread of misleading information about the disease through social media. Combatting COVID-19 misinformation on online platforms is crucial to the success of official authority efforts to control the pandemic. Manual fact-checking organizations may not be able to keep up with the misleading information that appears daily during the pandemic, creating a need for computational models capable of automatically identifying misinformation. Therefore, several research efforts have been dedicated to creating labeled misinformation datasets and utilizing them to develop and evaluate various ML and DL models. Most of these datasets have binary annotation in which the text (*e.g.*, tweet, post, *etc.*) is labeled as credible or non-credible. However, the misleading news stories related to COVID-19 are not equal, and some types of misleading information (*e.g.*, fake cure) could be life-threatening and should be instantly removed from social media platforms. For this reason, we've collected COVID-19-related Arabic tweet data containing 6,681 tweets. We manually annotated our data with fine-grained multi-label misinformation classes. Furthermore, we annotated a subset of this dataset (after removing the misleading tweets) with multi-class situational information classes. Identifying the situational information is very important for authorities and individuals to assess the situation during crises and understand the public's moods and needs. We performed several types of analysis to prove the quality of our annotations. Besides, we defined several classification tasks and applied multiple ML and transformer-based classifiers to train misinformation and situational information classification models. The experimental results confirm the validity of our data in developing different types of ML and DL models. The results also prove that AraBERT-COV19, a transformer-based model pretrained on large unsupervised COVID-19 Twitter data, outperformed the other methods by a significant margin with a micro-F1 score of 81.6% and a macro-F1 score of 73.4% for the multi-class misinformation and situational classification tasks, respectively. Regarding the multi-label misinformation task, the label power set using linear SVC as a base classifier outperformed the other tested methods with micro-F1 score of 76.69%. For future work, we could consider incorporating additional features related to author profiling and user engagement

to improve ML classifiers' performance instead of relying only on features extracted from the tweet contents. Another possible direction is to enrich our dataset versions and flatten their skewed class distribution by collecting more data, which greatly supports deploying the trained systems using this data in real-life applications.

### Funding
The authors received no funding for this work.

### Competing Interests
The authors declare that they have no competing interests.

### Author Contributions
- Rasha Obeidat conceived and designed the experiments, analyzed the data, performed the computation work, prepared figures and/or tables, authored or reviewed drafts of the article, data annotation quality analysis, and approved the final draft.
- Maram Gharaibeh conceived and designed the experiments, performed the experiments, analyzed the data, performed the computation work, prepared figures and/or tables, authored or reviewed drafts of the article, data annotation quality analysis, and approved the final draft.
- Malak Abdullah conceived and designed the experiments, prepared figures and/or tables, authored or reviewed drafts of the article, and approved the final draft.
- Yara Alharahsheh performed the experiments, analyzed the data, prepared figures and/or tables, data annotation quality analysis, and approved the final draft.

### Data Availability
The raw data and code are available in the Supplemental Files.

### Supplemental Information
Supplemental information for this article can be found online at http://dx.doi.org/10.7717/peerj-cs.1151#supplemental-information.

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
