# Peer review of "Multi-label multi-class COVID-19 Arabic Twitter dataset with fine-grained misinformation and situational information annotations"

_PeerJ Computer Science, doi:10.7717/peerj-cs.1151_

## Round 0.1 · original submission · Major Revisions

Dear authors,

Thank you for submitting your manuscript to PeerJ Computer Science.

We have completed the evaluation of your manuscript. The reviewers recommend reconsideration of your manuscript following major revision. I invite you to resubmit your manuscript after addressing the comments made by the reviewers. In particular:

1. Provide a better explanation of the choice of Arabic BERT models used for the comparison.

2. Fix spelling errors and review the grammar and writing style.

3. Remove the redundancies and unnecessary information mentioned by reviewer 2.

4. Systematically address all the suggestions from the three reviews and provide a point-to-point response in your rebuttal letter.

I hope you can complete the recommendation changes in a revision of your article.

Best,
Ana Maguitman


Reviewer 1 ·

Basic reporting

The authors propose a new dataset for covid19 misinformation classification with multi-label and multi-class labels that target a tweet's credibility and situational information awareness.
The topic was well presented with clear labels or class definitions. The dataset gathering methodology is unambiguous and well written.

Overall the paper is well structured and contains almost a sufficient amount of related works reviews and background information.

Experimental design

Although the experiments are all well designed with relevant comparison of sota deeplearning and machine learning algorithms, I'm questioning the choice of Arabic BERT models used for the comparison.
The authors chose AraBERTv02, arabert-c19 and mBERT. A better choice would have been AraBERTv02-twitter https://huggingface.co/aubmindlab/bert-base-arabertv02-twitter , MARBERT https://huggingface.co/UBC-NLP/MARBERTv2 in addition to the AraBERT-c19 model that they already tested on. Also for the multi-lingual model Twitter XLM-R https://huggingface.co/cardiffnlp/twitter-xlm-roberta-base is a better for comparison than mBERT since all the models that I mentioned are pre-trained on twitter data instead of MSA arabic data.

Validity of the findings

Results discussion showcases clearly where the models fail and the few weaknesses of the dataset proposed. The authors conclusion on why Arabert-c19 performs the best is also a valid outcome.

Additional comments

If you can try the improve the blurring of the twitter accounts shown.

There are some minor grammar mistakes to improve

Reviewer 2 ·

Basic reporting

I have several concerns and questions regarding the quality and clarity of writing. Please find my comments next, ordered by importance.
- The paragraph starting from the following sentence, isn't clear: "Since we depend on tweets that are already fact-checked into “False” or “True,” we did have to verify whether every tweet is credible (“True”) or not (i.e. “False”) except for particular credibility sub-classes.."
- "We think that the high-level binary(or ternary) classification doesn’t take into consideration fact-checking the worthiness of the tweet " This issue has been already studied and existing annotated Arabic, English, etc. Twitter datasets exist especially under the CheckThat! lab at CLEF in 2020 and 2021. I was surprised to see that Arabic datasets in this lab weren't cited in this paper.
- It wasn't clear how readers of the manuscript can acquire the dataset.
- The abstract should be made more concise and to the point. For example, the first few sentences could be summarized to a single sentence.
- The introduction should be majorly summarized to make it more concise.
- I suggest adding a sub-section on existing classification models to the related work section.
- I think a table summarizing and comparing the existing datasets will be very useful to understand the difference between these datasets and the proposed one
- The related work section should make the differences between the proposed datasets and existing ones very clear.
- Please correct the subfigures numbering in Figure 1 as the bottom two figures have the same literal "a".
- There's a presentation issue repeated in the manuscript. I think tables and figures should be clearly described and presented before the information in them discussed,
- I believe the section "Tweet Length" should be deleted or summarized considerably as it doesn't offer new/useful information about the data or differences between classes.
- "we don’t distinguish the primary class label from the additional class labels during modeling." This is an implementation detail that should be left for experimental evaluation section.
- There's no need to explain how the classical classification methods like SVM work. It is enough to motivate their use.
- Please have a full and thorough proof-reading pass as there are many grammatical and spelling issues. For example, some words are capitalized while they shouldn't be: "Fake" on line 103. Other sentences are completely mixed, e.g.: "most of the tweets have a The lengths of tweets range from 3 to 87 words"
- I highly recommend adding a working example in the introduction to differentiate between the different labels/classes identified.
- Discussion of hyper-parameter tuning is redundant across the first two sub-sections in the experimental evaluation section.

Experimental design

The manuscript presents the first multi-Label multi-class Twitter dataset for COVID-19 misinformation. The manuscript offers details on dataset construction and evaluation of multiple existing classification models over the dataset. Some essential justifications on design decisions are lacking. Below are further comments and questions to help the authors further improve the submission. The comments are ordered by their importance.
- Two concepts are repeatedly confused across the paper which are credibility and veracity. A claim made by a famous news agency (e.g., Aljazeera) might look "credible" to readers, but it might be actually conveying factually-incorrect information, i.e., it is false but credible. Please unify the discussion on one of these two concepts.
- Since data collection was done over multiple existing datasets, in addition to the method of collecting vaccine-related tweets, I believe the dataset might suffer from duplicate tweets which can be problematic during system training/testing. Kindly quantify and discuss duplication in the dataset (e.g., compute cosine similarity between all tweets pairs)
- Since the credibility classes and situational information classes are the main contributions of the work, the process of coming-up with these classes should be made much clearer.
- In Table 2, it seems the same "class" is once "true" and another time "false". I am very surprised to see this, I believe we should separate the truthfulness of the claim in the tweet from the actual topical class. Please justify the reason behind doing this class assignment method.
- Why did you opt to collect tweets from multiple labelled and unlabeled datasets? why not focus on a single dataset?
- Please justify this choice "We only consider that the tweets are originally labeled as true or unrelated, and we excluded the tweets with misinformation."
- "The label cardinally of ArCOV19-MLC is 1.23" Please clarify this sentence and motivate it.
- Line 118 "(except the part that has misinformation)" please clarify this statement.
- In the "true claim" example in figure 1, I think the annotation of both tweets isn't accurate. The "true claim" in the figure states that there is no evidence that COVID vaccine causes infertility, but it doesn't claim that the vaccine cause/doesn't cause infertility, i.e., it doesn't side by either of these two cases. This makes the tweets in the figure not stating actually the same claim or opposite to that claim.
- Please justify the reason behind using two agreement measures (Cohen's kappa and Krippendorff’s alpha)
- Since "Donation", "Help Seeking", and "Emotional Support" are almost non-existent in ArCOV19-Sit, is it meaningful to keep these classes in this taxonomy (consequently, the tweets)? Maybe it is better to merge these groups into a common group like "Seeking/offering Help"?
- "A misinformation Detection sample of 2000" --> How was the sampling done?
- Cohen's kappa agreement levels are reported as percentages. Percentages of what? I suggest you stick to how Kappa values are reported usually (between -1 and 1)
- Please justify the reason behind verifying annotation quality using 1.5K tweets? Why not 1K or 2K for example?
- "recruited a second annotator" --> did the second annotator have access to the labels of the first annotator?

Validity of the findings

no comment

Additional comments

no comment

·

Basic reporting

• The article described by the researchers deals with a very hot and interesting topic.

• The depended language level is acceptable, but contains grammar errors. After rechecking these errors we can say that the manuscript is clear and unambiguous, professional English used throughout.

• The introduction section needs to be expanded to clarify different forms of Misleading Information. However, the manuscript has no individual background Theory section, so the introduction should cover it too. I believe that authors need to make sure what kind of misleading information thy are targeting (misinformation, disinformation, or malinformation) and the have to add a suitable reference(s) to show the difference between them; many researchers are misusing the misinformation term and wrongly using it to describe all kings of Misleading Information.

• The related work section is well organized, and new references are depended. You need to support some of the addressed related works by numerical results. The numerical results of previous works will help you make a fair comparison between your results on one side and those of the previous works on the other side.


• There is no doubt Misleading Information achieved high popularity due to the pandemic. This leads to various attackers finding sophisticated methods to vulnerate systems from different aspects, e.g., economically to gain profits, politically to destroy either countries or public figures, etc . In general, my appreciation is that the article indeed has a very detailed work regarding challenges when annotating data. The use of Fine-grained Credibility and Situational Information Annotations along side with machine learning classifiers make this work attractive to any data analysis researcher. However, I did not find a discussion regarding what impact the research described in the article has on health professionals and decision makers. I miss discussion regarding who is interested in using these sophisticated techniques, who is the main interested party, and how the proposed technique by researchers improves information quality, among others.


• The article structure is acceptable. The figures are relevant to the article's content and appropriately described and labelled. Titles of the figures have been put below the figures correctly. But unfortunately, for tables, their titles also been put below them, please remove them to be above the tables. The data well represented in the tables. All Figures and Tables are referred in the manuscript text sufficiently.

Positive aspects of the investigation:

- The dataset used in the investigation is an outstanding contribution. I congratulate the article's authors for exposing these datasets that can eventually be used in other research.
- The methodological description is well detailed, specifically in the description of the approach.


• I have a comment on the description of different machine learning algorithms, no need to provide such a description of a very well known algorithms. You name algorithms that you are utilizing and give a reference or more, I prefer a reference book or a book chapter, to readers who want to learn more about them!

Experimental design

There is a section of the article called "experimental setup" and another one called "experimental setup", but IHMO, I don't see a discussion of the results anywhere. I recognize that the experiments' inputs are well described, but I don't know the lessons learned from the experiments. On the other hand, I have seen several papers that expose the results of machine learning techniques in different areas of knowledge. The authors of this article describe the results very well. But, this makes me ask, "so what?". Let me explain, it is fine to describe the results regarding the application of novel techniques, but I do not see what impact these results have in ​​detecting Misleading Information. For example, what is the point of knowing that Random Forest has better results when it comes to multiclass credibility classification? There is no doubt that this is a significant result for the Misleading Information detection context. But, for ​​your dataset’s and system’s end users, what use is this data? These questions are essential when convincing a group of governmental authorities to finance your research, for example, security mechanisms in sensitive institutions.

Validity of the findings

Another aspect that makes me strange is the word "experiments". Reporting an experiment, in general, involves the description of subjects, hypotheses, description to evaluate the hypotheses, the experimental design, among others. I believe that what the authors report is a quasi-experiment or an analytical study on machine learning techniques in the field of misleading information detection. But, perhaps someone very purist in experimentation can refute the methodological process used in the article.

• You browsed good results . Still, you only discussed the results without producing suitable recommendations to the reader to support your findings. Please focus on your aims and contributions that give your outcomes more weight. Hence, the impact and novelty will be assessed and be meaningful replication encouraged where rationale & benefit to literature is clearly stated.

• The obtained data provide a good plan to compare them with previous works but need to be explained clearly and combined with the research gab.

• Conclusions are well stated, linked to the original research question & limited to supporting results. It is necessary to add critical numerical outcomes to the Abstract and Conclusions sections. You need to suggest some points for future work as an extension to this work. Please give straightforward suggestions.

Additional comments

Additional comments
Finally, I suggest the authors discuss the impact of the research results in the context of misleading information and their applications in one section. What contribution does this research have to different parities? Who benefits from the results of this research in the area of fighting against misleading information?

---

## Round 0.2 · Minor Revisions

The three reviewers are happy with the revised version of your manuscript. Once you address the two minor issues indicated by reviewer 1 your manuscript will be ready for publication.

Reviewer 1 ·

Basic reporting

The authors addressed my comments as well as the other reviewers' comments.

This version of paper is a sizable improvement of the previous one. Language is clearer and unambiguous. Experiment designs and result reporting are relevant and well presented. The addition of the detailed literature comparison highlights well the authors contributions.

Two minor things need to be fixed:
- The citation for Twitter XLM-R (you can call it XLM-T) should be for Barbieri et al. first in addition to Conneau et al. Bibtex: https://huggingface.co/cardiffnlp/twitter-xlm-roberta-base
- line 618, AraBERT instead of ArBERT, since the Marbert authors also released a model called ArBERT

Experimental design

no comment

Validity of the findings

no comment

Reviewer 2 ·

Basic reporting

First, I would like to thank the authors for their detailed replies to all my concerns. I specifically admire (and appreciate!) the effort put by the authors to address my comment about duplicates (Comment 18 for reviewer #2) as it shows the level of dedication of the authors to re-do almost everything based on this change. I also appreciate that they have addressed many of my comments in the manuscript. I believe the manuscript is generally ready for publication.

Experimental design

I think the manuscript sufficiently described the dataset collection method and the experimental setup.

Validity of the findings

No comments here.

·

Basic reporting

I believe the authors have made a good effort to address my concerns, and the article can be considered for publication in its current form!

Experimental design

Reviewed before!

Validity of the findings

Reviewed before!

Additional comments

Congratulations to authors for their great job!

---

## Round 0.3 · accepted · Accept

Congratulations! Your manuscript has been accepted for publication. Thanks for your contribution to PeerJ Computer Science.